# A Device Designed to Improve Care and Wayfinding Assistance for Elders with Dementia

**Winger Seiwo Tseng \* and Jonny Fang**

Graduate School of Industrial Design, National Yunlin University of Science and Technology, Douliou 640, Yunlin, Taiwan
\* Correspondence: tsengws@yuntech.edu.tw; Tel.: +886-972-727-193

**Abstract:** With an aging population and the rapid increase in the rate of dementia, the care of patients is a significant problem for caregivers and family members at home. Patients' spatial and environmental cognitive impairments require caregivers to pay attention to their walking routes, walking safety, and avoiding dangerous areas. With this in mind, this study developed the Dementia Care Management and Mentoring Guarantee System (DECMGSS) to examine the accuracy and efficiency of patient wayfinding, and to reduce the stress on institutional caregivers through a wayfinding task and a caregiver stress scale (CSS). The results showed that the mean time spent with the DECMGSS on 10 subjects with mild to moderate dementia was significantly less than that without the device ($t = -2.930$, $p = 0.017$), and the number of errors was also less but not significantly different. In addition, the DECMGSS did significantly reduce the caregiver stress load. Twenty institutional caregivers were divided into experimental and control groups. There was no significant difference in the stress load scores of the control group before and after the test. However, a significant mean difference was found between the pre-test and post-test scores of the experimental group ($t = 3.315$, $p < 0.009$). DECMGSS's intervention significantly reduced the caregiver's personal anxiety and stress in caregiving and the patient's time dependence on the caregiver. Although this product is primarily used for patients in the Care Center, it can also be used in other home care settings.

**Keywords:** spatial cognitive impairment; care management; guidance security; wayfinding for dementia; service experience engineer

## 1. Introduction

There are more than 50 million patients with dementia in the world. With the aging of society and the extension of life expectancy, the number of people suffering from dementia will continue to rise [1]. In 2018, Taiwan had more than 280,000 people with dementia, and it is estimated that in 2061, there will be more than 880,000 people with dementia. The cognitive ability of patients with dementia is degraded; they cannot handle life affairs independently, and they are highly dependent on the assistance of caregivers [2]. Dementia has a long disease course, which places significant pressure on caregivers [3]. Compared with the care of elderly patients without dementia, the care of family members with dementia needs the provision of more assistance and safety monitoring in daily life. Therefore, these caregivers have more emotional, financial, and physical stress than ordinary caregivers.

Dementia is a major neurocognitive disorder [4], which makes those living with dementia unable to take care of themselves in their daily life, and produces behaviors and symptoms of dementia. Patients have wandering behaviors, depression, delusions, hallucinations, restlessness, sleep disorders, and other behaviors that cause significant impacts on their lives and the people around them [5]. Wandering is the second most frequent behavioral problem in dementia [6]. One of the causes of wandering is that the patients' perception of space and memory are impaired [7]. They are unable to perform correct planning for their own behavior [8]. The wandering behavior of patients is one

of the leading causes of care difficulties. However, the cognitive impairment of spatial orientation makes patients often become lost in their living environment [9] and even go outside of the scope of care. When wandering, there is a high risk of accidental injuries such as falls and fractures [10]. These injuries caused by wandering behaviors require caregivers to pay extra attention to the status of dementia patients in the environment, in addition to their own work. Therefore, the safety of patients' daily behaviors virtually causes the potential psychological pressure and burden for caregivers at work.

## 1.1. Spatial Cognitive Impairment in Dementia

Wandering is a common behavior of patients with dementia, and it easily causes a burden on caregivers. There are many possible reasons for wandering [11], among which spatial disorientation makes patients unable to position themselves in the correct position in the environment. Spatial orientation disorder is also a symptom of early-stage Alzheimer's, which will cause the patients to lose their autonomy [9]. It is also one of the reasons for serious care difficulties and human burdens. Spatial cognitive impairment leads to the difficulty of wayfinding for patients with dementia and severely impacts their life. For example, patients may suffer from incontinence and emotional distress because they cannot find the location of the toilet or cannot find the place of the room, resulting in an impact on the rest of the time. The deterioration of their wayfinding ability and the development of spatial cognition of patients with dementia present a reverse form, which are followed by the spatial cognitive abnormalities of "other center" and "self center", and finally, the recognition of landmarks [12].

From the study of the relationship between the location of brain injury lesions and cognitive impairment in humans, Aguirre [13] combined animal experiments to classify spatial cognitive impairment into four categories:

1. Landmark agnosia: The lesion is located in the posterior parietal lobe of the brain, and the cognitive impairment is that the person no longer responds to a building or feature initially used as a landmark. This is a form of visual agnosia.
2. Egocentric agnosia: The lesion is located in the posterior parietal lobe of the brain, and cognitive impairment is the person's inability to represent the location of objects from the perspective of the self. The patient can recognize landmarks and their placement in space but cannot walk through the streets to reach their destination. This patient has lost the ability for "egocentric route learning".
3. Heading agnosia: Patients with heading agnosia are unable to walk to the target, although they know it is there, and cannot form a sense of direction in the external environment.
4. Anterograde agnosia: The location of the brain injury, including the right hippocampus and parahippocampal gyrus (PHG), is cognitively impaired in that the patient does not become lost in a "familiar" environment [14] but has no learning ability in an "unfamiliar" environment. The ability to form memories of the learning environment is lost.

The development of spatial cognition in humans is firstly, the recognition of landmarks, and secondly, the construction of route knowledge. Paths are related to landmarks, and they continue to develop, and the connected landmarks and paths form a collection. Finally, a coherent frame of reference is developed between and across clusters, and finally, investigative knowledge is formed [15]. This shows that the cognitive process of human exploration of the environment is hierarchical. These hierarchical landmarks and paths form the Skeletal Structure framework [16].

Therefore, compensatory information from the environment that provides important landmarks and information to the patient with dementia can help the patient easily establish landmark learning and recognition. Important landmarks and environmental information should be placed at key reference points, such as path bifurcations and the patient's personal space, where pathfinding problems are likely to occur so that patients can easily adopt path

learning and build self-centered cognitive maps that can more effectively help patients solve their wayfinding problems [17].

### 1.2. Care for Dementia

Care institutions provide full-time and reasonable care to avoid the aggravation of disability [18]. Dementia care is complicated and labor-intensive. The contents of their care work include helping residents take a bath, helping them move around, changing clothes, turning over, helping them eat, paying attention to water intake and medication distribution, and recording their heartbeat and blood pressure [19]. At the same time, the caregiver must also be responsible for the cleaning and disinfection of the environment, housekeeping, replacement of bed sheets, confirmation of the safe fixation of the bed wheel and rail, cleaning of residents' clothes, confirmation of the smoothness of the moving path, and other non-care-work contents [19]. While carrying out general affairs, it is still necessary to pay attention to the living care needs of developmentally disabled residents in institutions, or to whether there are emergencies that need to be handled. This often causes caregivers to feel busy at work or affects their rest time [20]. In general, patients with mild or moderate disabilities can move on their own, which is often one of the reasons for the difficulty of care. For example, caregivers often need to pay attention to the safety of the patient's walking around when they perform general care work. Sometimes, spatial cognitive impairment leads to dementia patients having incontinence and emotional distress because they cannot find the location of the toilet or cannot find the location of the room, which affects the rest time. Therefore, the difficulty of wayfinding can only be completed with the help of caregivers [21,22]. This extra work is constant and difficult to master, forming a potential psychological burden and work pressure for caregivers [23]. The care of dementia is very important for the families of people with dementia and can also be said to be the last line of defense for care. Tseng et al. [22] summarized the potential deficiencies and needs in maintenance institutions through semi-structured interview and context observation analysis to help patients find their way and aid in care management: (1) landmark setting; (2) user-centered guidance service; (3) clearance of objects in the moving path of the environment; (4) mobile safety monitoring of patients in the environment; (5) interface interaction mode of text, voice, and image; and (6) ancillary services that do not affect the patient's actions. Through the above unmet needs, it is necessary to develop home care management and indoor guidance service systems, combined with the Internet of Things technology, to help patients safely reach the desired target space indoors, and to monitor and record patients' daily behaviors, and reduce the human burden of care in institutions.

### 1.3. Technology-Assisted Care

Many studies have aimed to improve or address the cognitive impairment of patients with dementia. Most were aimed at the method of identifying and making decisions on the patient's travel path. Liu et al. [24] used PDA to navigate patients with cognitive impairment, and they found that text display was more efficient than images. If words were combined with sound prompts, the efficiency of guidance was improved. Reagan and Baldwin [25] believed that rotating the sound, distance, orientation, and image would improve the accuracy of the guidance and reduce the reaction time. If the entity landmark image was combined with the direction indicator, the error rate was the lowest, and the platform was easy to use [26]. In addition, if landmarks were presented with meaningful and familiar objects [27], the probability of patients returning by themselves was improved. Grierson et al. [28] used the belt as a guiding tool. Most patients with dementia could turn correctly according to the belt. However, if the amount of navigation information was simplified, it can effectively improve the accuracy of patients' wayfinding. Qianzhongyi [29] found that the accuracy of patient-centered navigation was the same as that of normal subjects. Still, if the floor was changed (stairs or elevators), it was easy for the patient to become lost.

However, the above techniques can guide patients, and through the data and records of positioning and trajectory tracking, we can know the location of dementia patients in real-time. Through the records of trajectory, we can also analyze their behavior and judge their wandering behavior. Megges et al. [30] studied the user experience of using a mobile device application (APP) and a GPS positioning watch on the market as positioning assistance technology. Fu et al. [31] used infrared rays in an indoor space to detect the movement trajectories of the elderly to analyze their wandering behaviors and judge the symptoms of dementia. Huang and Wang [32] used infrared and Bluetooth to judge the situation of leaving the bed and wandering in the indoor environment. Lee and Chuang [33] calculated position positioning based on signals from mobile devices and base stations. They used acceleration sensors for fall detection and fall state analysis for elderly and dementia patients. Tseng et al. used a DW1000 follower UWB chip plus Esp-01s Wi-Fi as the equipment for an indoor positioning system, which could achieve a centimeter-level accurate positioning effect [22].

Based on the above literature, it can be seen that the impairment of spatial cognition in patients with dementia will lead to the patient's inability to reach their goals [34] and abnormal behaviors such as impaired road recognition, wandering, getting lost, or even losing their way [35]. These behaviors make caregivers pay more attention to the patient's walking status [36,37], the safety of walking [38], and preventing the patient from entering dangerous areas or leaving the care institution. For the caregivers of institutions, in addition to their daily care and cleaning work, also need to care for many elderly people at the same time. The wandering behaviors of these patients are sources of significant work stress and burdens.

Based on these factors, the Dementia Elderly Care Management and Guidance Security System (DECMGSS) has been developed and combined with Esp-01s Wi-Fi signal determination and a UWB chip module to achieve the function of indoor positioning with high accuracy. It improves the accuracy and efficiency of wayfinding for patients with dementia and reduces the pressure on institutional caregivers to care for patients with dementia.

## 2. Materials and Methods

In this study, two experimental tasks, the wayfinding task and the Caregiver Stress Scale, were used to verify that the DECMGSS can improve the wayfinding efficiency of dementia patients using a wearable device and landmark interaction system. With the indoor positioning system and Monitoring App, the caregivers can learn about the patient's behavior and the current situation in the environment at any time through the monitoring screen and record their essential daily activities through the app, such as drinking water, going to the toilet, eating, and rest and social interaction, to reduce the work pressure of caring for patients with dementia.

The study was conducted in accordance with the Declaration of Helsinki, and the protocol was approved by the Human Research Ethics Committee at National Chung Cheng University (CCUREC 110072101) on 23 September 2021.

### 2.1. Hypothesis

**Hypothesis 1.** *Patients using the DECMGSS can reduce the wayfinding time and improve their accuracy.*

**Hypothesis 2.** *Institutional caregivers using the DECMGSS can reduce the stress value of caring for patients with dementia.*

### 2.2. Stimulus

2.2.1. Dementia Elderly Care Management and Guidance Security System (DECMGSS)

The DECMGSS was designed based on our previous insight study (Tseng et al., published in 2021) [22], which observed the dementia patients and caregivers in a nursing center and summarized six potential unmet needs: (1) offering indoor user-centered guidance, (2) providing the instant location information of elders with dementia to caregivers,

(3) landmarks setting, (4) assistance notification, (5) environmental route planning, and (6) use of a wearable device as a guide for indoor route guidance [22]. It includes the following three subsystems and one wearable device: (1) the Indoor Guidance system; (2) the Guiding Landmark and Interactive system; and (3) the Monitoring system with App [22] (Figure 1).

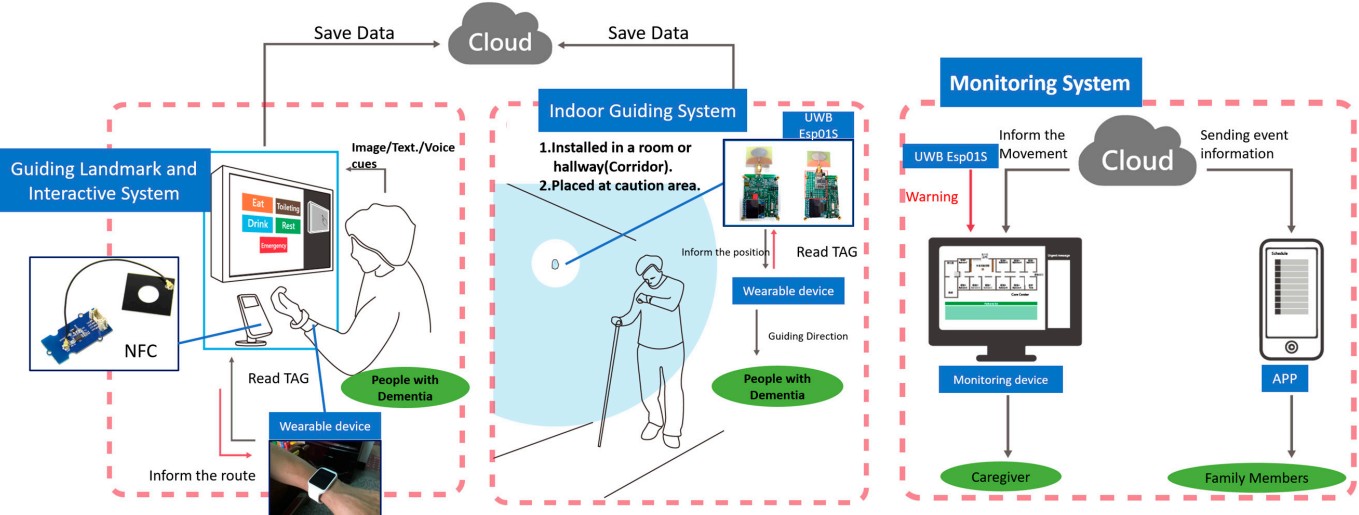

**Figure 1.** Dementia Elderly Care Management and Guidance Security System, including the Indoor Guidance system, the Guiding Landmark and Interactive system, the Monitoring system, and the user operation diagram [22].

1.    Indoor Guidance System

This Indoor Guidance system (IGS) uses the Esp-01s anchor of the ultra-wideband (UWB) system, installed indoors to detect the smartwatch, and the Esp-01s tag, worn by dementia patients. It records the data returned by patients from the tag and the anchor and uses the time difference of arrival (TDOA) [39] to position the patients indoors. Landmark interactive devices will be set at the places the residents regularly go to every day, such as restaurants, tea rooms, toilets, friendship halls, and their bedrooms, for eating, drinking, toileting, watching TV, talking, and taking a rest. The NFC signal of the smartwatch is connected with the landmark interactive system. Each resident has a different code and sets the destination through the voice, image, or text information set on the landmark. The nearfield communication element of the wearable device will use the orientation sensor to judge the destination and current position, obtain the guidance direction (arrow), calculate the path with the current position, and then guide with the display direction of the wearable device screen. When the patient moves around the destination, the UWB tag wearer will sense the Esp-01s signal. When the signal meets the destination, the smartwatch will terminate the guidance and display the word "arrival". When residents complete these five behaviors, such as eating, drinking, toileting, watching TV, talking, and resting, the safety monitoring system and the app will record the completion times for caregivers to query.

During the guidance process, the patient must wear a device to display the direction. Therefore, the built-in sensing module of the wearer must include a Bluetooth module and a Wi-Fi module as the communication function and the acceleration sensing module as the sensing function during steering. In this study, the Sony Smartwatch 3 was selected as the functional carrier of wearable devices (Figure 2).

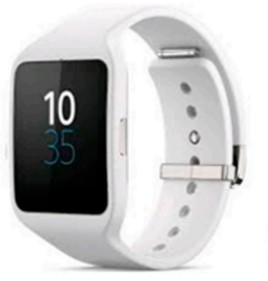 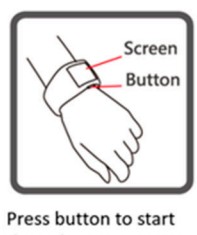 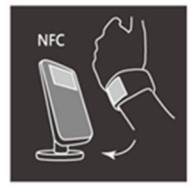 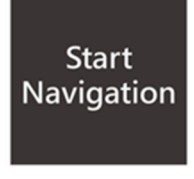 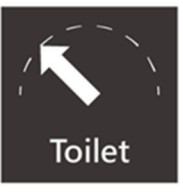 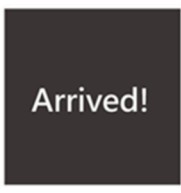

**Figure 2.** The wearable device of the Sony Smartwatch 3 and the guidance operating process.

2. Guiding Landmark and Interactive System

The Guiding Landmark system and the interactive device aim to improve the wayfinding ability of residents with dementia in the indoor environment and reduce caregivers' care burden. The guiding landmark is composed of a touch screen and an NFC sensor. The touch screen is erected on the wall about 150 cm above the ground to facilitate the residents with dementia in operating and viewing the wayfinding information. The erection site is at the turning point of the path in the maintenance institution, the door of the residential room, the toilet, the tea room, the friendship hall, the restaurant, and other frequently used space positions. The screen interface displays daily activities for the patient to choose from (Figure 3). The patient first connects to the NFC with a wearable device. Touching the screen interface will display text and pattern options for eating, drinking, resting, and going to the toilet, and asking "what do you want to do" by voice. After the patient uses a voice or touch pattern to choose an activity, the guiding landmark will transmit the value to the wearer and guide him to the destination. The wearable device will sense the UWB anchor signal when the patient arrives at the destination (Figure 4). When the signal meets the destination, the smartwatch will terminate the guidance and display the word "arrival". When the smartwatch completes an operation, the system will use Wi-Fi as the transmission medium to back up the interactive records and the operation time in the cloud for the monitoring system to view the data. The smartwatch then returns to its initial state.

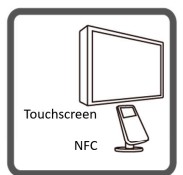 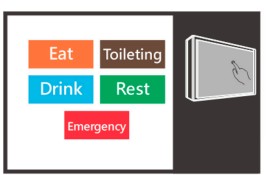 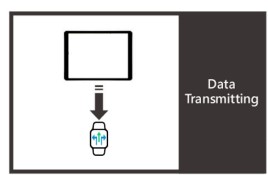
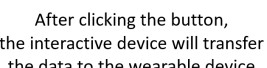 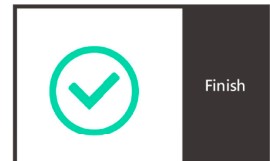

**Figure 3.** Operation diagram of Guiding Landmark and Interactive device.

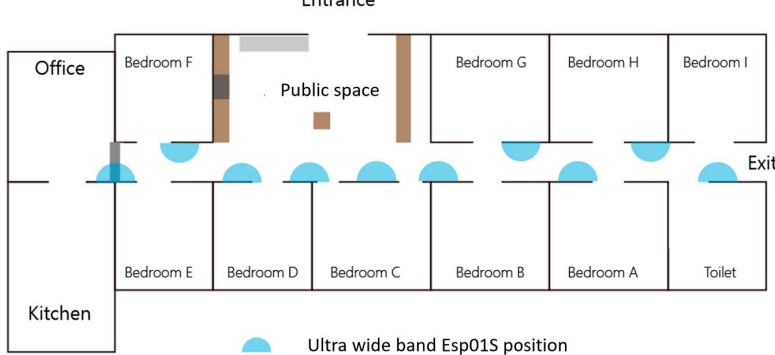

**Figure 4.** Examples of the ultra-wide band's position settings in the Care Center [18].

## 3. Monitoring System App

The Monitoring system App is based on two potential demands, according to Tseng et al.'s research [22]: (1) locating the close location of dementia patients and guiding, as well as monitoring the patient to the destination, and (2) recording the critical events of their daily life (going to the toilet, drinking water, eating, resting, social interaction, etc.). On the right side of the screen is the emergency notice bar. When a sensor set in a dangerous area detects a patient, it will directly display this and remind the caregiver.

In this study, the three-axis accelerometer built into the Sony Smartwatch 3 was used to perform fall detection on patients. The signal magnitude vector (SVM) proposed by Mathie et al. and Karantonis et al. [40,41] was used to calculate the vector strength G-value formula (1). When SVM > 1.8 g, the patient is at high risk of falling. However, the G-values of rapid sitting and backseat falls, or rapid lying and backseat falls in daily life (Activity Daily life), are very similar and cannot be judged from SVM alone. In this case, the patient's signal magnitude area (SMA) (2) formula should be used to determine their activity level. When the SMA exceeds a threshold value within a 1 s interval, the patient is active; otherwise, the patient is stationary. With the g-value returns from SVM, the Z-axis acceleration value $\alpha\_Z$ is used to calculate the angle between the human posture and the gravitational force $\varnothing$, ($\varnothing$ = arc cos($\alpha z$)) formula (3), when $\varnothing$ > 80, it is lying down, and the *x*- and *y*-axis acceleration values are used to determine that it is lying, lying on the side, or lying down, and other different postures, and send out the danger signal for falling. Participants were classified as upright if their tilt angle was between 0 and 60°, while values of between 60 and 120° indicated a recumbent position; for any greater tilt angle, participants were classified as inverted.

The equations used in this study to detect falls were as follows:

$$SVM = \sqrt{(A_x)^2 + (A_y)^2 + (A_z)^2} \tag{1}$$

$A_x$, $A_y$, and $A_z$ denote acceleration (g) in the *x*-, *y*-, and *z*-axes.

$$SMA = \frac{1}{t}\left( \int_0^t |x(t)|dt + \int_0^t |y(t)|dt + \int_0^t |z(t)|dt \right) \tag{2}$$

x(t), y(t), and z(t) refer to the body components of the *x*-, *y*-, and *z*-axis samples, respectively.

$$\varnothing = arc\cos(\alpha z) \tag{3}$$

The user's tilt angle ($\Phi$) is defined as the angle between the positive z-axis and the gravitational vector g.

The safety monitoring app can accurately locate the location of residents and judge the status of residents as being active, resting, searching, or in need of help, as shown in Table 1 and Figure A1. The need for help can be divided into three types according to the urgency of the situation: emergency (immediate notice), need for help, and general notice (Table 1). Through the UWB device, the safety monitoring system will display the overall indoor map of the care environment and the number of people in each area (bedroom, corridor, and activity space). At the bottom of the indoor map, a table showing the name of the living room and the living patient will be displayed. If the patient stays in the room, the name column indicates green; otherwise, it is gray, which is intended to inform the caregiver whether the patient stays in the room (Figure A1).

### 2.2.2. Caregiver Stress Scale (CSS)

The CSS used in this study is based on Feng [42] and Kuo et al.'s questionnaire [43] and is designed for nurses and caregivers in long-term institutions. It includes 22 questions on four domains: personal response, job concern, job competence, and inability to perform private work. The rest of the questions were not used. They are different from the target population of this study because they belong to home-based dementia care. The final CSS

questionnaire used in this study consisted of four scales: eight questions on "Personal response", six questions on "Job concern", five questions on "Job competence", and five questions on "Time dependence" (Table A1). Each question was scored on a five-point Likert scale, with each question being scored 1–5, with higher scores indicating higher levels of work stress.

**Table 1.** The monitoring app can determine the status of a resident with dementia and present information to notify caregivers according to the level of urgency of the need for help.

| Classification | | Definition | Follow-Up | Color Marking |
|---|---|---|---|---|
| **Activity in progress** | | Detects the location of dementia patients as participating in institutional care activities or classes, such as gymnastics, classes, etc. | Displayed and recorded on monitors and smartphones | **GREEN** |
| **Resting** | | The signal of dementia patients is detected in their own bedroom during the rest time of the institution. | Displayed and recorded on monitor and smartphones | **GREEN** |
| **Wayfinding** | | The person with dementia has selected the desired activity and is being directed to the target field. | Real-time tracking is displayed and recorded on the monitor and smartphone. | **GREEN** |
| **Need help** | | A caregiver must be notified when a person with dementia has the following behaviors. | | |
| | Immediate notification | Notify the caregiver immediately when the patient has the following conditions or behaviors: fall, the patient enters a dangerous area, such as the pool, kitchen, the door of the institution, etc. | Notify all caregivers urgently and deal with them immediately | **RED** |
| | Need help | 1. Inconsistent with the travel direction of the target field; 2. Stay where you are for 1 min; 3. Patients directly press the help notice; 4. Stay in the bedroom longer than the rest period. | 1. Inform the caregiver, 2. Use smartwatches for voice communication. | **ORANGE** |
| | General notice | When patients successfully arrive at the destination, complete the expected daily activities, such as drinking water, going to the toilet, eating, resting, etc. | Record the number and time, and display it on the caregiver's monitoring screen and the smartphone | **GREEN** |

### 2.3. Subjects

The subjects recruited for this experiment belonged to two private nursing facilities in Yunlin County, were divided into experimental and control groups and were assigned to participate in two experimental tasks.

1.   Subjects of the Wayfinding task:

For the experimental group, 10 subjects were recruited based on the following criteria: (1) 65 years of age or older, regardless of gender; (2) mild to moderate dementia, as measured using the Min-Mental Scale Examination; (3) good mobility, able to move around the room independently; (4) no visual impairment, no aggressive behavior; (5) able to communicate through Chinese or Taiwanese. A total of 15 nursing center residents were recruited for this experiment, and after testing, 5 were moderately severe, so only 10 subjects were accepted. Three males and seven females were present, with an average age of 72 years old, ranging from 68 to 93 years old; five residents were not literate. Scores on the MMSE scale ranged from 15 to 24, which falls within the range of cognitive ability for mild

to moderate dementia, which was consistent with the conditions of the subjects in this experiment. After participating in the physical landmark task (control task), all participants had a one-week experience with the "Demented elderly care management and guidance security system" (DECMGSS) and the "Indoor Interactive Landmark Device" wayfinding task experiment (experimental task).

2. Subjects of the Caregiver Stress Scale Task:

The 20 caregivers enrolled in this experiment were 2 males and 18 females, aged between 33 and 62, with the majority of caregivers aged 40–49. (1) The primary caregiver of a resident with dementia, (2) at least one year of caregiving experience at the facility, and (3) the ability to communicate through Chinese or Taiwanese. The 10 caregivers in the experimental group and the 10 caregivers in the control group were used to compare the change in the Caregiving Stress Scale before and after the intervention of the Dementia Care Management and Guidance Monitoring Safety System. The caregivers in the experimental group were the caregivers of subjects in the two phases of the "Wayfinding Task". The experimental groups of subjects and caregivers also experienced the "Indoor landmark interactive system and wearable device" for 8 h a day for one week, while the caregivers experienced and operated the Guided Monitoring System APP.

### 2.4. Wayfinding Task Experiment

In the wayfinding task experiment, all subjects had to perform two wayfinding tasks, the control task and the experimental task. For the latter task, the subjects had to wear a smartwatch and a UWB positioning tag and complete the wayfinding task through the indoor interactive landmark device, smartphone, and UWB positioning guidance (Figure 5). In the physical landmark task, the subjects wore no auxiliary products. They were only guided to their destinations by setting up physical landmark pictures (showing the images of toilets, pantries, social halls, etc.) and directional guidance at each corner.

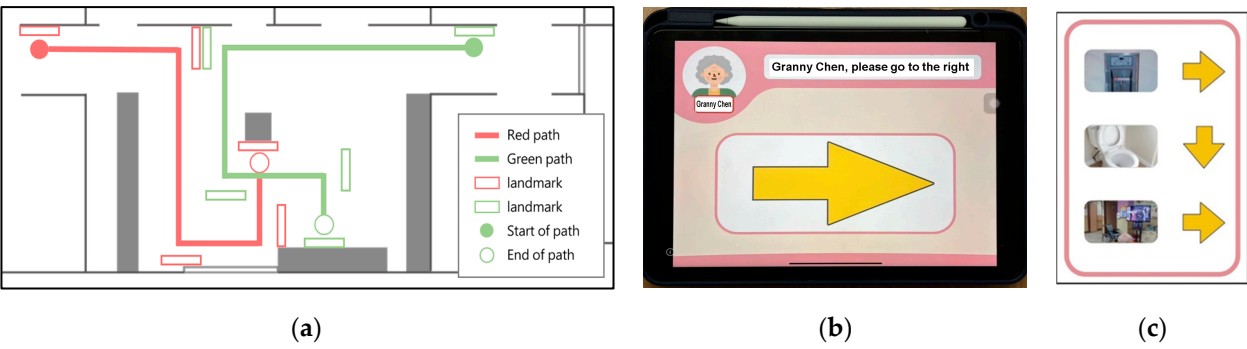

(**a**)          (**b**)          (**c**)

**Figure 5.** (**a**) Route planning map of two wayfinding tasks, (**b**) the indoor interactive landmark device, and (**c**) a schematic diagram of the physical landmark.

The "physical landmarks" were presented in the same way as the landmarks in public spaces, where multiple locations were shown on a single landmark at the same time, and the participants needed to judge the destination and direction they wanted to go. The route planning of the wayfinding task experiment is shown in Figure 5, with red and green paths, both 32 m in length and three turns, and the indoor interactive landmark devices or physical landmarks installed at the turns of the paths according to the wayfinding task (experiment or control group). All participants were required to complete the experiment and control wayfinding tasks. The two paths were randomly assigned to two tasks. If the first path was path Green, the next task was path Red, which prevented the subjects from having any practice effect.

- Process for the wayfinding experiment: 1. After the subject's check-in, he/she and his/her guardian were first explained the experiment and signed the informed consent form. 2. The MMSE test was conducted to determine the subjects with mild to

moderate cognitive impairment and eligibility for this experiment. 3. The wayfinding task was first explained. The physical landmark wayfinding task was decided by drawing lots for the Green or Red path, and the subject was then asked to complete the physical landmark wayfinding task. 4. The experiment manager fixed the smartwatch and UWB TAG on the subject's arm and matched the indoor interactive landmarks for operation and wayfinding instructions. Each patient and the caregiver had previously conducted the indoor interactive landmark guidance experience for one week. 5. One week after the control task, the wayfinding task of the indoor interactive landmark was conducted. 6. After the wayfinding task, a semi-structured interview was conducted to understand the usability of the indoor interactive landmarks for the wayfinding and guidance monitoring security system with the smartwatch and UWB tag.

- Coding method: The overall wayfinding time and the number of errors of the subjects in the two sets of tasks were recorded.

### 2.5. Caregiver Stress Scale (CSS) Experiment

The subjects of the CSS experiment were recruited from two private caregiving institutions in Douliu City, Yunlin County, and 10 groups were recruited from each institution to serve as the experimental and control groups. Caregivers in the experimental and control groups were tested twice on the CSS at one-week intervals as a pre- and post-test of CSS values. However, the caregivers in the experimental group were tested on the Caregiving Stress Scale before experiencing the DECMGSS and then again after one week of experience. In the experimental group, the caregivers experienced the DECMGSS and Guided Monitoring Security app for institutional care tasks. The residents they cared for would also wear smartphones and UWB Tags for Guided Monitoring and record their behavior for one week, 8 h per day. In the control group, the caregivers cared for the residents with dementia in the same way but without the DECMGSS system.

### 3. Results

### 3.1. Wayfinding Task

This study hypothesized that the Dementia Care Management and Guidance Monitoring and Safety System (DECMGSS) would reduce the wayfinding time and errors of patients with dementia. The time difference and number of errors between the experimental group (guided by the UWB positioning device and the indoor interactive landmark system) and the control group (guided by physical landmarks) were compared for the two wayfinding tasks. Table 2 shows that the average wayfinding time without any assistance was 130.70 s, and the average number of errors was 1.6. The average wayfinding time through the DECMGSS was 106.10 s, and the average number of errors made by the subjects was 1.4.

**Table 2.** Paired sample statistics of the average wayfinding time and the number of errors between the experimental and control groups in the two wayfinding tasks.

|  |  | Mean | N | Std. Deviation | Std. Error Mean |
|---|---|---|---|---|---|
| With the DECMGSS assistance | Wayfinding Time (s) | 106.10 | 10 | 40.698 | 12.870 |
|  | Number of errors | 1.40 | 10 | 0.71 | 0.22 |
| Without the DECMGSS assistance | Wayfinding Time (s) | 130.70 | 10 | 51.571 | 16.308 |
|  | Number of errors | 1.60 | 10 | 0.84 | 0.27 |

Regarding the time spent on the wayfinding task, we can see that t = −2.930, Sig. $p = 0.017$, which is a significant level rejecting the null hypothesis (Table 3). This indicates that the time spent by the subjects using the DECMGSS was significantly less than the time spent on the wayfinding task without this equipment. Therefore, it is inferred that residents

with mild to moderate dementia have better wayfinding performance with the assistance of this system compared to the general physical landmarks. In addition, the number of errors between the two tasks was not significantly different. However, there were fewer errors with the guidance of the wearable system and the indoor interactive landmarks.

**Table 3.** The *t*-test results of paired samples for time and the number of errors in the two wayfinding tasks.

| | Mean | Std. Deviation | Std. Error Mean | 95% Confidence Interval of the Difference | | t | d.f. | Sig. (2-Tailed) |
|---|---|---|---|---|---|---|---|---|
| | | | | Lower | Upper | | | |
| Two wayfinding tasks compared with and without the CMGSS's assistance | −24.60 | 26.55 | 8.40 | −43.60 | −5.60 | −2.93 | 9 | 0.017 |
| Number of errors between two wayfinding tasks | −0.20 | 1.13 | 0.36 | −1.01 | 0.61 | −0.55 | 9 | 0.591 |

However, since the number of participants was small, we further analyzed the effect size (ES) to understand the impact of DECMGSS on the pathfinding task. Additionally, the effect sizes of the two pathfinding tasks with and without the DECMGSS's assistance mean scores were tested, including Cohen's d = 0.526, with an effect size = 0.5. An effect size (ES) of 0.5 indicates that the mean with DECMGSS's assistance is at the 69th percentile without DECMGSS's help, with a non-overlap of 33% in the two distributions, which represents that DECMGSS's assistance had a medium-effect magnitude [44].

### 3.2. The Results of Caregiver Stress Scale Test

In this study, reliability analyses were conducted on 96 items of the pre-test and post-test for the experimental and control groups of the Caregiver Stress Scale test. Reliability is the degree of correlation between the observed and true scores. When the correlation between the observed and true scores is higher, it also reflects the consistency of the measurement results. The higher the consistency, the better the reliability of the questionnaire. Since this study used a five-point Likert scale, the internal consistency reliability, Cronbach's $\alpha$, was used, and its standard is 0.7 or higher [45]. The results showed that the internal consistency coefficients of Cronbach's alpha were 0.852, demonstrating acceptable reliability (Table 4).

**Table 4.** Reliability statistics of Caregiver Stress Scale Test.

| Reliability Statistics | |
|---|---|
| Cronbach's Alpha | N of Items |
| 0.852 | 96 |

The mean pre-test and post-test CSS scores for the experimental group were 68.4 and 64.7, respectively, and the difference between them was 3.7. The difference between the pre-test and post-test mean CSS scores for the control group was 1.7. The experimental and control groups showed a decreasing trend in their CSS scores in the post-test. However, the mean post-test total score was still higher for the experimental group than the control group. Still, the decrease was more significant for the experimental group, and the mean difference between the two groups was smaller.

In this study, a paired sample t-test analysis was conducted to assess whether the intervention of DECMGSS affected the CSS's measure. The results showed a significant mean difference between the pre-test and post-test scores of the experimental group (t = 3.31, $p < 0.009$). However, no significant difference was found between the control

group's pre-test and post-test mean scores (t = 0.781, $p < 0.455$), so it was concluded that the intervention of the DECMGSS could reduce the CSS values. Therefore, it is inferred that the intervention of the DECMGSS can reduce the caregiver's stress burden (Table 5).

**Table 5.** The *t*-test analysis of the difference between the pre-test–post-test samples of the experimental and control groups of the Caregiver Stress Scale.

| Pre-Test vs. Post-Test Analysis | Mean Dif. | Std. Deviation | Mean Value of Std. Error | t | Degree of Freedom | Sig. (2-Tailed) |
|---|---|---|---|---|---|---|
| Experimental group | 3.70 | 3.52 | 1.11 | 3.31 | 9 | 0.009 |
| Control group | 1.70 | 6.88 | 2.17 | 0.78 | 9 | 0.455 |
| Sub-configuration of Experimental group | | | | | | |
| Personal response | 3.00 | 4.11 | 1.30 | 2.30 | 9 | 0.046 |
| Job concern | −0.60 | 1.57 | 0.49 | −1.20 | 9 | 0.260 |
| Job competence | −0.70 | 1.63 | 0.51 | −1.35 | 9 | 0.209 |
| Time dependence | 2.00 | 2.44 | 0.77 | 2.58 | 9 | 0.030 |

We further examined the experimental group's effect size (ES) to analyze the benefits of the DECMGSS on caregiver stress burden. Additionally, the effect sizes of the pre-test and post-test of the experiment group's mean scores were tested, including Cohen's d = 0.37, with an effect size = 0.3. An effect size (ES) of 0.3 indicates that the mean of the post-test is at the 62nd percentile of the pre-test, with a non-overlap of 21.3% in the two distributions, representing that the DECMGSS's assistance had a low effect magnitude [44].

We further analyzed the experimental group's four sub-dimensions of the nursing work stress scale. We found there was a significant difference between the intervention of this product in the aspects of personal response (t = 2.3, $p < 0.046$ *) and time dependence (t = 2.58, $p < 0.030$ *). Based on the analysis results, it is concluded that the intervention of the DECMGSS helps reduce the caregivers' personal response and the time-dependent two-dimensional scale of CSS (Table 5). Among the eight sub-questions of personal response, we found significant differences in the pre- and post-tests for "I feel anxious about taking care of residents "(t = 3.0, $p < 0.015$ *) and "I feel nervous because of taking care of residents" (t = 2.45, $p < 0.037$ *) (Table A1). In other words, the use of the DECMGSS significantly reduced anxiety and nervousness in the caregiver's personal response to caring for the residents. From the five sub-tests of time dependence, there was a significant difference in the pre- and post-test scores of "I need to keep an eye on the patient" (t = 2.75, $p < 0.022$ *) and "Taking care of the patient gives me no time to rest" (t = 3.67, $p < 0.005$ *), indicating that the application of this product allows the caregiver to have more time to manage their time (Table A1).

## 4. Discussion

### 4.1. Enhancing the Effectiveness of Wayfinding Performance

The indoor interactive landmark device developed in this study can be described as a combination of audio, text display, and pictorial guidance [24]; it is combined with the ESP-01S-tag wearable device and the Sony smartwatch [22] to directly guide patients with dementia to the target location. The time spent on the wayfinding performance shows that the average wayfinding performance of the "indoor interactive landmark device" is significantly faster than when using the general "physical landmarks" to complete the wayfinding task. This echoes the study of Reagan and Baldwin [25], where the rotation of sound, distance, orientation, and image was found to improve the accuracy of the guidance and reduce the response time. In addition, the "indoor interactive landmark device" combines the findings of Kaminoyama et al. [26] at the turn of the path, combining landmark and directional finger guidance, which also allows patients with dementia to

quickly determine the direction in which they should walk. The indoor interactive landmark devices serve as compensatory information from the environment during wayfinding. Illustrative interactive directions such as audio, textual displays, and graphic animations can reduce the burden of judgment on dementia residents and quickly guide them to build landmark knowledge and establish self-centered path maps [17].

In addition, the number of errors between the two tasks was not significantly different. Two points can be explored. The first is that the guided tasks of indoor interactive landmarks were set up with new landmarks, and their guided indoor target paths were also new settings, while the physical landmarks were traditional landmark displays, which required learning to generate familiarity and construct path knowledge [16], while physical landmarks do not. The second indoor interactive landmark task design should be adjusted to allow participants to first familiarize themselves with the exterior form of each landmark and the location of the landmark and gradually construct knowledge [16] of the path to the landmark through a hierarchical process of cognitive exploration of the environment [17], and finally conduct a guided task. Therefore, this task design should be included in the experimental tasks in future research.

*4.2. Effectiveness of Reducing the Burden of Caregiving Stress*

The dementia elderly care management and guidance security system (DECMGSS) enables caregivers to more easily grasp the dynamics and status of residents with dementia, responding to the needs of "recording and sharing care information of residents with dementia" and "grasping the status of residents with dementia at any time", as in the previous study [22]. In this study, the care-work stress scale was used to assess whether DECMGSS helps reduce caregivers' nursing workload and was effective in reducing the overall nursing work stress scores. This can be inferred from the fact that DECMGSS can help caregivers to understand the daily behaviors of people with dementia and guide them to where they want to go because without this potential intangible burden [23], they can focus on and manage their tasks, such as the content of caregiving (refer to lines 103–106), described by Tseng et al. [19]. This was particularly evident in the personal reaction and the time dependence. In particular, the items "anxiety about caring for the residents" and "nervousness about caring for the residents" were significantly reduced in the caregivers. Almost all of the time-dependent items showed significant differences, indicating that the system is effective in helping caregivers to manage time and to interact with patients. In addition, this system ensures that caregivers do not have to pay attention to patients at all times and can have time to rest; this also echoes the study of Chen et al. [23]. However, the system did not achieve significant differences in work competency and work concern. In future planning and design, it is necessary to increase the existing work management of caregivers to reduce their work stress.

*4.3. Research Limitations*

In this study, the Dementia Elderly Care Management and Guidance Security System (DECMGSS) was developed and tested using the Wayfinding Task and the Caregiver Stress Scale Test. It improved the reduction in wayfinding time and stress care values for patients with dementia. However, the experimental design of the pathfinding task should include an indoor interactive landmark memory task to enable the participants to construct path knowledge, reduce the number of errors and improve the efficiency of pathfinding.

The results obtained so far are limited by the fact that the number of prototypes was only one, which resulted in an insufficient sampling of the number of subjects. Therefore, large-scale clinical trials are expected to be conducted to solidify the study results.

This study uses contextual inquiry insights (previously published journal) to identify potential deficiencies and needs in nursing care facilities and develops the dementia elderly care management system and guidance security system, which include (1) the Indoor Guidance system, (2) the Guiding Landmark and Interactive system, and (3) the Monitoring system with App [22]. Through the Indoor Guidance system, the Guiding Landmark and

Interactive system, and the smartwatch with a UWB location device for patients with dementia, the patients can be guided to their actual target locations, such as toilets, pantries, restaurants, social rooms, and their own rooms, and these movements are recorded through the Monitoring system with App. The system can record essential daily behaviors (such as drinking, toileting, eating, and going out) and uploads them to the cloud. Through the real-time location of patients' information and the indoor landmark interactive devices, the Monitoring App can determine the state and environment of the patient's behavior, whether immediate help is needed to reduce the caregiver's burden, and record and compare each daily behavior.

Finally, the dementia elderly care management and guidance security system was found to be effective in guiding patients with mild to moderate dementia to their destinations, spending significantly less time on average and making fewer errors than those who did not use the device. However, the error difference was not significant. In addition, the "Dementia Elderly Care Management and Guidance Monitoring Safety System" did significantly reduce caregivers' caregiving burdens; it also significantly reduced caregivers' anxiety and stress in caregiving and significantly reduced the patient's time dependence on the caregiver. Although this product is primarily used for patients in the Yunlin County Care Center, it can also be used in other home care settings.

The system will be considered for outdoor wayfinding in the future, but the outdoor wayfinding technology will require adding a GPS positioning system. The outdoor guidance system is targeted at the elderly living alone and at home and is in line with the government's community long-term care program. Landmark systems will be installed at community facilities, such as community centers, shared restaurants, sports and recreational parks, and individual home gateways, so that seniors can wear a wayfinding system to guide them back to their homes after a shared lunch at noon. If they want to exercise or go for a walk, they can also go to the community park through the system, making their retirement life more relaxing and less burdensome to their families.

**Author Contributions:** All authors contributed part to the paper. J.F. collected and organized data, as well as the programming. W.S.T. conducted conceptualization, methodology, validation, formal analysis, wrote the manuscript and acted as a corresponding author. All authors have read and agreed to the published version of the manuscript.

**Funding:** This research was funded by the Ministry of Science and Technology, grant number MOST 110-2221-E-224-037.

**Institutional Review Board Statement:** The study was conducted in accordance with the Declaration of Helsinki, and the protocol was approved by the Human Research Ethics Committee at National Chung Cheng University (CCUREC 110072101), 23 September 2021.

**Informed Consent Statement:** Informed consent was obtained from all subjects involved in the study.

**Conflicts of Interest:** The authors declare no conflict of interest.

## Appendix A

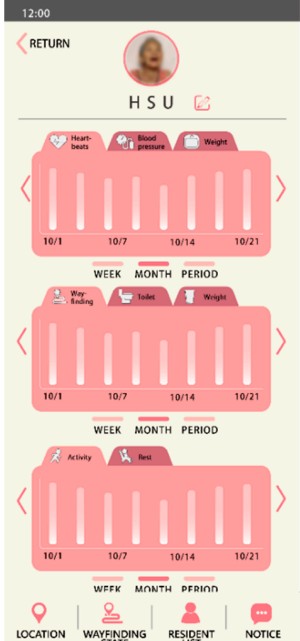

Interface of monthly records of daily behavior.

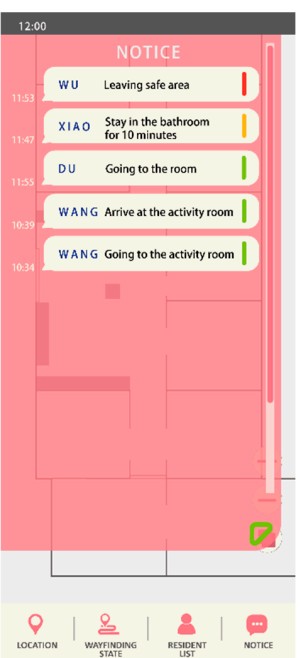

Resident list: Patient daily event input interface.

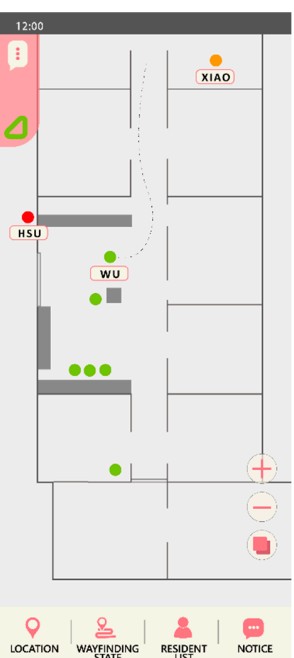

The patient's wayfinding status in the spatial location.

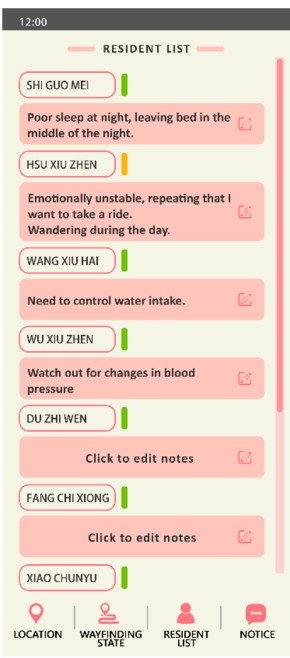

Notice: The resident's immediate behavioral status, to inform Caregivers based on the level of urgency of the need for assistance.

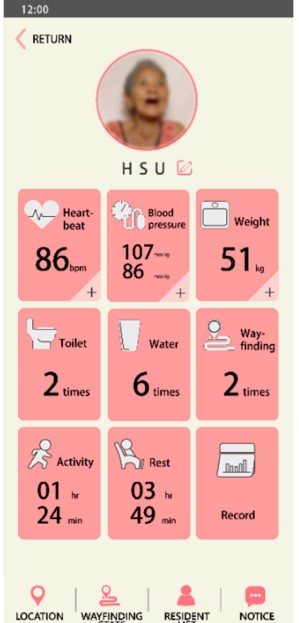

Daily essential behavior record of Patient Hsu.

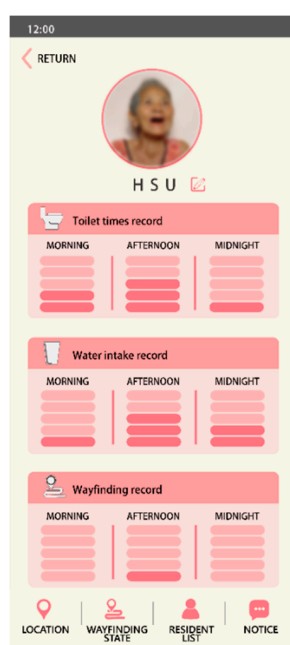

Patient Hsu's daily essential behavior record.

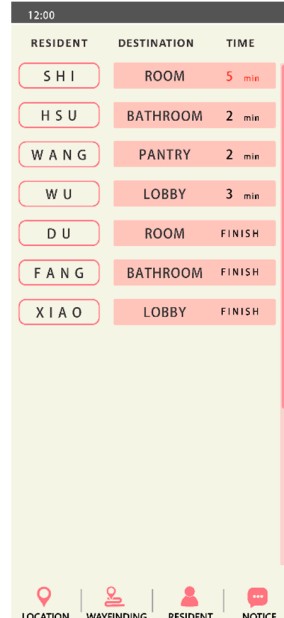

List of all patients' wayfinding results.

**Figure A1.** Monitoring APP—Care daily event logging and management interface.

**Table A1.** The *t*-test analysis of paired samples of the main and sub-dimensions of the Caregiver Stress Scale questionnaire in the experimental group.

| | Pre-Test vs. Post-Test Analysis | Mean | Std. Deviation | t | Sig. (2-Tailed) |
|---|---|---|---|---|---|
| **Personal response** | I have felt frustrated in taking care of residents | 0.200 | 1.033 | 0.61 | 0.555 |
| | I feel anxious about taking care of residents | 1.000 | 1.054 | 3.00 | 0.015 * |
| | I sometimes feel angry when I am with residents | 0.200 | 1.229 | 0.51 | 0.619 |
| | I feel emotionally unstable because of taking care of residents | 0.300 | 0.823 | 1.15 | 0.279 |
| | I feel nervous because of taking care of residents | 0.800 | 1.033 | 2.44 | 0.037 * |
| | I have been impatient when taking care of residents | 0.100 | 0.568 | 0.55 | 0.591 |
| | I have felt helpless in caring for residents | 0.200 | 1.033 | 0.61 | 0.555 |
| | I have trouble sleeping because of caring for the residents | 0.200 | 0.919 | 0.68 | 0.509 |
| | The overall analysis of Personal Response | 3.00 | 4.11 | 2.30 | 0.046 * |
| **Job concern** | Worried about the deterioration of the resident's condition | −0.100 | 0.568 | −0.55 | 0.591 |
| | Unable to answer questions from residents or their families | 0.000 | 0.667 | 0.000 | 1.000 |
| | Unable to meet the needs of the residents | −0.600 | 0.516 | −3.674 | 0.005 * |
| | Feel uncomfortable discussing problems with residents or family members | 0.000 | 0.667 | 0.000 | 1.000 |
| | Not having enough time to do what needs to be done | 0.000 | 0.943 | 0.000 | 1.000 |
| | Feeling behind in caring for the residents | 0.100 | 0.568 | 0.557 | 0.591 |
| | The overall analysis of Job Concerns | −0.60 | 1.57 | −1.20 | 0.260 |
| **Job Competence** | The institution has given me more work than I am capable of doing | −0.200 | 0.919 | −0.688 | 0.509 |
| | My job does not allow me to have full autonomy | −0.200 | 0.422 | −1.500 | 0.168 |
| | I am unable to work with my colleagues | −0.100 | 0.876 | −0.361 | 0.726 |
| | I am not able to do my job satisfactorily within my capabilities | −0.100 | 1.101 | −0.287 | 0.780 |
| | I do not feel confident | −0.100 | 0.568 | −0.557 | 0.591 |
| | The overall analysis of Job Competence | −0.70 | 1.63 | −1.35 | 0.209 |
| **Time dependence** | I need to help patients with many daily tasks | 0.600 | 0.843 | 2.250 | 0.051 |
| | The patient is dependent on me | 0.400 | 0.843 | 1.500 | 0.168 |
| | I need to keep an eye on the patient | 0.800 | 0.919 | 2.753 | 0.022 |
| | I need to help the patient with many life skills that he has forgotten about | −0.400 | 0.699 | −1.809 | 0.104 |
| | Taking care of patients leaves me no time to rest | 0.600 | 0.516 | 3.674 | 0.005 |
| | The overall analysis of time-dependence | 2.00 | 2.44 | 2.58 | 0.030 |

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
