# Peer review of "A Device Designed to Improve Care and Wayfinding Assistance for Elders with Dementia"

_sustainability, doi:10.3390/su141711076_

Round 1

Reviewer 1 Report

Subject is of interest and attention should be paid for home devices to improve elderly life condition.

Statement is confusing, please rephrase: " However, there was no significant difference in the control group. DECMGSS's intervention significantly reduced the caregiver's personal anxiety and stress in caregiving and also significantly reduced the patient's time dependence on the caregiver." (lines 19-22).

The introductory part should be more outlined from previous works, maybe some medical facts added to the definition of dementia patients. For example, there are only spatial (wayfinding) disorders in dementia patients or are there also cognitive disorders (relative to what the elders wanted to do, not only the location to arrive)? 

Please explain better the necessity of designing a Wi-Fi system been designed for indoor orientation "which can achieve the function of indoor positioning with high accuracy" (line 147-148), as, according to Table2, the number of errors has not been significant with the DECMGSS’s assistance. Maybe a larger number of experiments can outline the benefits of the method? 

Please explain more detailed how dangerous situations (such as falls) can be detected by the system.

Can the proposed system be employed also for outdoor orientation, where the probability of getting lost is more acute?

What do the minus-signs in Table 3 stand for?

Explain  more clearly what the Cronbach's alpha coefficient implies (line 363).

More studies should be made to asses the obtained results. 

The conclusion section should be extensive, to explain better the benefits of the proposed system.

The conclusion section should be extended

Author Response

Response to Reviewer 1 Comments

We are very grateful to the reviewer1’s comments which make the results of this paper more solid. The manuscript was revised in accordance with the reviewers' comments and suggestions, followed by professional extensive English editing (English language editing by MDP, Attachment 1). Followings are our description on revision according to the reviewer1’s comments.

  1. Abstract: The English text of the abstract was re-edited and partially rewritten. (Please refer to 1. Introduction, Line17-22)
  2. Introduction: The English text of the introduction was re-edited and examined, and new literature on spatial cognitive impairment with dementia was added, and the redundant words and phrases were also revised (please refer to 1. Introduction, line 69-101).
  3. research methods: In this paper, we add a detailed description of fall detection techniques (see lines 268-301)
  4. Results: In the results section, we recruited too small a sample of subjects, so we further investigated the statistical differences and actual effect sizes (see Lines 435-440, Lines 469-475.).
  5. Discussion: We have rewritten the paper based on the previous literature and added new study limitations. (Refer to Line 494-555.)

6.The concluding section has been adjusted. Please refer to line 582-591

Comments and Suggestions for Authors:

Point 1: Subject is of interest and attention should be paid for home devices to improve elderly life condition.

Response 1: Thank you very much for your comments. Research and development of home products for the dementia community will help the elderly to get more help.

Point 2: Statement is confusing, please rephrase: " However, there was no significant difference in the control group. DECMGSS's intervention significantly reduced the caregiver's personal anxiety and stress in caregiving and also significantly reduced the patient's time dependence on the caregiver." (lines 19-22).

Response 2: Thanks to the reviewers for the reminder. Please refer to Line 18-21.

Point 3: The introductory part should be more outlined from previous works, maybe some medical facts added to the definition of dementia patients. For example, there are only spatial (wayfinding) disorders in dementia patients or are there also cognitive disorders (relative to what the elders wanted to do, not only the location to arrive)?

Response 3:Thank you very much for your comments. Please refer to Introductory section: Line 69-101.

Point 4: Please explain better the necessity of designing a Wi-Fi system been designed for indoor orientation "which can achieve the function of indoor positioning with high accuracy" (line 147-148)

Response 4: Thank you very much for your comments. Has been explained more clearly please refer to Line 200-205

Point 5: as, according to Table2, the number of errors has not been significant with the DECMGSS’s assistance. Maybe a larger number of experiments can outline the benefits of the method?

Response 5: Thanks to the reviewers' comments, we should make sure that the experimental design of the wayfinding task should be adjusted. Please refer to the Research limitations section: line 544-555, and 4.1. Enhancing the effectiveness of wayfinding performance line 512-522.

Point 6: Please explain more detailed how dangerous situations (such as falls) can be detected by the system.

Response 6: Indeed the presentation of this part should be clearer, for fall detection, please refer to Section. 3.Monitoring System App, line 268-301.

Point 7: Can the proposed system be employed also for outdoor orientation, where the probability of getting lost is more acute?

Response7: Thanks to the reviewer's suggestion, this expanded idea has been put in the conclusion, please refer to line 544-555.

Point 8: What do the minus-signs in Table 3 stand for?

Response8: The "-" sign appears in Table 3 because it is the post-task score minus the pre-task score, and the pre-task time is longer, so it is a negative sign.

Point 8: Explain more clearly what the Cronbach's alpha coefficient implies (line 363).

Response 8: Thanks to the reviewers' comments, please refer to section 3.2 line441-450.

Point 9: More studies should be made to asses the obtained results.

Response 9: We thank the reviewers for their comments. The results obtained so far are limited by the fact that the number of prototypes was only one, which resulted in an insufficient number of subjects being sampled. Please refer to Line 495-555

Point 10: The conclusion section should be extensive, to explain better the benefits of the proposed system.

Response 10: The concluding section has been adjusted. Please refer to line582-591

Reviewer 2 Report

The manuscript entitled “A Device Designed to Improve Care and Wayfinding Assistance for Dementia Elders” presents an experimental study aimed to determine the effectiveness of a technological device over the care and wayfinding assistance for elders with dementia. The topic is of great social and clinical interest, with relevant practical implications. The paper is well written and presented. The authors include all the relevant information in a way that is easily understood by the reader. The study is methodologically sound, and the findings are clearly presented and discussed. I only have three points that should be addressed by the authors:

1.       The authors should provide a brief rationale about the sample size calculation of the study and about its power and confidence implications.

2.       The discussion section should be significantly expanded. The authors did a great job with the rationale of the study, but the discussion lacks a proper discussion with previous studies.

3.       In addition, a Limitation subsection is advised, specially as the study was performed with a very limited sample size.

Author Response

Response to Reviewer 2 Comments

We are very grateful to reviewer2’s comments which make the results of this paper more solid. The manuscript was revised in accordance with the reviewers' comments and suggestions, followed by professional extensive English editing (English language editing by MDP,). Followings are our description on revision according to the reviewer2’s comments.

This paper was rewritten and integrated the introduction with the literature discussion. At the same time, the research methods/results/discussions/conclusions are also readjusted.

Comments and Suggestions for Authors:

The manuscript entitled “A Device Designed to Improve Care and Wayfinding Assistance for Dementia Elders” presents an experimental study aimed to determine the effectiveness of a technological device over the care and wayfinding assistance for elders with dementia. The topic is of great social and clinical interest, with relevant practical implications. The paper is well written and presented. The authors include all the relevant information in a way that is easily understood by the reader. The study is methodologically sound, and the findings are clearly presented and discussed. I only have three points that should be addressed by the authors:

  1. Abstract: The English text of the abstract was re-edited and partially rewritten. (Please refer to 1. Introduction, Line17-22)
  2. Introduction: The English text of the introduction was re-edited and examined, and new literature on spatial cognitive impairment with dementia was added, and the redundant words and phrases were also revised (please refer to 1. Introduction, line 69-101).
  3. research methods: In this paper, we add a detailed description of fall detection techniques (see lines 268-301)
  4. Results: In the results section, we recruited too small a sample of subjects, so we further investigated the statistical differences and actual effect sizes (see Lines 435-440, Lines 469-475.).
  5. Discussion: We have rewritten the paper based on the previous literature and added new study limitations. (Refer to Line 494-555.)

6. The concluding section has been adjusted. Please refer to lines 582-591

Point 1: The authors should provide a brief rationale about the sample size calculation of the study and about its power and confidence implications.

Response 1: Thank you very much for your comments. Since there was only one system and prototype developed in this study, and the COV-19 epidemic coincided with it, the sample of subjects recruited was too small, so we further investigated the statistical differences and the actual effect size.

Please refer to Line435-440, Line469-475.

Point 2: The discussion section should be significantly expanded. The authors did a great job with the rationale of the study, but the discussion lacks a proper discussion with previous studies.

Response 2: The discussion section has been revised. Please refer to Line495-543.

Point 3: In addition, a Limitation subsection is advised, specially as the study was performed with a very limited sample size.

Response 3: The Limitation section has been added. Please refer to Line544-555.

Reviewer 3 Report

Line 44-45. The same sentence is repeated.

Line: 359. It would be interesting to indicate what are the most common errors made in the two wayfinding tasks, of what type, for the purpose of recording for future research.

Author Response

Response to Reviewer 3 Comments

We are very grateful to reviewer3’s comments which make the results of this paper more solid. The manuscript was revised in accordance with the reviewers' comments and suggestions, followed by professional extensive English editing (English language editing by MDP,). Followings are our description on revision according to the reviewer3’s comments.

  1. Abstract: The English text of the abstract was re-edited and partially rewritten. (Please refer to 1. Introduction, Line17-22)
  2. Introduction: The English text of the introduction was re-edited and examined, and new literature on spatial cognitive impairment with dementia was added, and the redundant words and phrases were also revised (please refer to 1. Introduction, line 69-101).
  3. research methods: In this paper, we add a detailed description of fall detection techniques (see lines 268-301)
  4. Results: In the results section, we recruited too small a sample of subjects, so we further investigated the statistical differences and actual effect sizes (see Lines 435-440, Lines 469-475.).
  5. Discussion: We have rewritten the paper based on the previous literature and added new study limitations. (Refer to Line 494-555.)

6.The concluding section has been adjusted. Please refer to line 582-591

Comments and Suggestions for Authors:

Point 1: The same sentence is repeated.

Response 1: Repeated sentences have been deleted. Please refer to Line45-46.

Point 2: It would be interesting to indicate what are the most common errors made in the two wayfinding tasks, of what type, for the purpose of recording for future research.

Response 2: Thanks to the reviewers for the reminder. However, this error type was not included in the study because the sample was very small and did not produce many errors. However, in the next clinical study with a larger sample, we will definitely include this error type in the study because it will be of great benefit to the improvement of the system.

Reviewer 4 Report

General comments:

-          The paper is well-written and presented in a well-structured manner.

-          The paper is scientifically sound and the experimental design appropriate to test the hypothesis.

-         -                  If possible, summarize the related work into the tabulation with pros and cons, so the reader can easily find the gap in research in published work  and get an idea what you contribute here.

-          If possible, compare the results with existing methods with the help of figure or table.

-          Discuss the limitations of the developed approach.

Author Response

Response to Reviewer 4 Comments

We are very grateful to the reviewer4’s comments which make the results of this paper more solid. The manuscript was revised in accordance with the reviewers' comments and suggestions, followed by professional extensive English editing (English language editing by MDP, Attachment 1). Followings are our description on revision according to the reviewer4’s comments.

  1. Abstract: The English text of the abstract was re-edited and partially rewritten. (Please refer to 1. Introduction, Line17-22)
  2. Introduction: The English text of the introduction was re-edited and examined, and new literature on spatial cognitive impairment with dementia was added, and the redundant words and phrases were also revised (please refer to 1. Introduction, line 69-101).
  3. research methods: In this paper, we add a detailed description of fall detection techniques (see lines 268-301)
  4. Results: In the results section, we recruited too small a sample of subjects, so we further investigated the statistical differences and actual effect sizes (see Lines 435-440, Lines 469-475.).
  5. Discussion: We have rewritten the paper based on the previous literature and added new study limitations. (Refer to Line 494-555.)

6.The concluding section has been adjusted. Please refer to line 582-591

Comments and Suggestions for Authors:

Point 1: The paper is well-written and presented in a well-structured manner.

Response 1: Thank you very much for your comments.

Point 2: The paper is scientifically sound and the experimental design appropriate to test the hypothesis.

Response 2: Thank you very much for your comments.

Point 3: If possible, summarize the related work into the tabulation with pros and cons, so the reader can easily find the gap in research in published work and get an idea what you contribute here.

Response 3: Thank you very much for your comments. However, the results and contributions of the research are presented in the form of manuscript writing.

Point 4: If possible, compare the results with existing methods with the help of figure or table.

Response 4: Thank you very much for your comments. However, the results and contributions of the research are presented in the form of manuscript writing.

Point 5: Discuss the limitations of the developed approach.

Response 5: Thank you very much for your comments. The Limitation section has been added. Please refer to Line541-551.